# Impact of the Timing of Maternal Peripartum Depression on Infant Social and Emotional Development at 18 Months

**DOI:** 10.3390/jcm11236919

**Published:** 2022-11-23

**Authors:** Jaqueline Wendland, Xavier Benarous, Héloïse Young, Takoua Brahim, Gisèle Apter, Nicolas Bodeau, David Cohen, Priscille Gérardin

**Affiliations:** 1Psychopathology and Health Processes Laboratory, Institute of Psychology, University Paris Cité, 92774 Boulogne Billancourt, France; 2Department of Child and Adolescent Psychiatry, Pitié-Salpêtrière University Hospital, Assistance Publique-Hôpitaux de Paris, 75013 Paris, France; 3Department of Child and Adolescent Psychopathology, Amiens University Hospital, 80000 Amiens, France; 4Research Group for Analysis of the Multimodal Cerebral Function, Institut National de la Santé et de la Recherche Médicale (INSERM U1105), University of Picardy Jules Verne (UPJV), 80025 Amiens, France; 5Child and Adolescent Psychiatric Department, Rouen University Hospital, University Rouen Normandie, 76031 Rouen, France; 6Child and Perinatal Psychiatric Department, Le Havre University Hospital, University Rouen Normandie, 76600 Le Havre, France; 7CNRS UMR 7222, Institute for Intelligent Systems and Robotics, Sorbonne University, 75007 Paris, France; 8Centre de Recherches sur les Fonctionnements et Dysfonctionnements Psychologiques (CRFDP, EA 7475), 76821 Mont Saint Aignan, France

**Keywords:** depression, anxiety, pregnancy, postnatal, child outcomes

## Abstract

The study assessed how the timing of maternal perinatal depressive symptoms affects infant socio-emotional characteristics at age 18 months. The study was a longitudinal cohort study that included six assessment points from the third trimester of pregnancy up to age 18 months (±1 month). Assessment of mothers included the Edinburgh Postnatal Depression Scale and the State-Trait Anxiety Inventory, while assessments of infant included the Infant Toddler Social and Emotional Assessment (ITSEA) at 18 months. Mothers were categorized into one of the following groups: mothers who presented postnatal depression only (*n* = 19); mothers who presented both prenatal and postnatal depression (*n* = 14), and mothers who never showed perinatal depression symptoms (*n* = 38). Mothers who presented both prenatal and postnatal depression showed significantly higher levels of depressive score, reactivity to stress and level of anxiety trait compared to mothers of the two other groups. Infants of prenatally and postnatally depressed mothers had higher scores on the internalizing subscore of the ITSEA. The number of depression episodes during the study period was positively correlated with the externalizing and internalizing subscores of the ITSEA. These findings support the need to provide specific screening to identify women with prenatal depression.

## 1. Introduction

Perinatal depression and anxiety symptoms represent a major public mental health issue, affecting at least one in 10 women during pregnancy or in the year after birth [1,2,3]. These conditions expose the mother to a wide range of personal, family and social burdens, and represent lifetime health-related costs for society [4]. Beyond increased obstetric risks, early exposure to maternal depression and anxiety may also affect the development of the fetus and the child. These conditions predispose the child to physiological and neurological alterations, probably via epigenetic mechanisms that regulate gene activity, neurobiology, and behavior, and contribute to shape the infant’s early socioemotional skills [5,6,7,8]. This acquired vulnerability may then increase the risk of an array of unfavorable outcomes in children, in particular poor cognitive, nutritional, social, and emotional development [9,10].

The recognition of the potential impact of peripartum depression on women and children has led to growing attention towards the course of women’s mental health problems across the perinatal period and beyond. Longitudinal cohort data with multiple assessments in pre- and postnatal periods were used to document the heterogeneity within the trajectories of peripartum depressive symptoms and associated impairments in infants [11,12,13,14]. Ultimately, these studies showed that the severity and the chronicity of maternal peripartum depressive symptoms were both key risk factors for cognitive and emotional difficulties in infants [15].

How the timing of maternal peripartum symptoms (i.e., prenatal/postnatal period) moderates this relationship is, however, difficult to determine, with contradictory findings according to considered outcomes. The period of exposure to maternal internalizing symptoms may be critical for the acquisition of specific skills but not others [16,17]. Findings from studies conducted to determine the links between maternal peripartum depressive symptoms and infant neurodevelopmental outcomes (e.g., cognitive function, language) [18,19,20] cannot be directly extrapolated to the infant’s emotional development.

Another gap in the literature concerns the effect of associated anxiety symptoms in children of depressed mothers. Previous studies support the assumption that anxiety has an independent effect on infant development during the prenatal period, while its specific effect is less well documented in the postnatal period [21,22]. Again, longitudinal studies can help distinguish the impact of acute/reactional anxiety (e.g., worries related to pregnancy) from severe and possibly impairing forms of maternal anxiety. Few studies have been designed to disentangle the separate prenatal and postnatal effects of anxiety on infants. In addition, other risk factors for infant development often co-occur, leading to complex interpretation in terms of causal relationship. These issues are, however, worth determining, as the mechanisms involved, the impact on infants, and the potential interventions may differ depending on the periods considered [23].

The current study tried to address some of the limitations mentioned above. As a follow-up to a previous assessment at 12 months [24], the main objective of the present study was to examine prospectively the impact of three maternal perinatal depression profiles in low-risk pregnant women on infant social-emotional characteristics at ages 12 and 18 months. In particular, we sought to examine whether infants whose mothers presented pre- and postnatal depression had different social-emotional characteristics than those whose mothers showed postnatal depression only, as compared to those whose mothers never showed depression symptoms from pregnancy to 18 months. We also assessed the links between the number of maternal depression episodes throughout the study and infant social-emotional characteristics at 18 months, as well as the association between maternal prenatal trait and state anxiety symptoms and infant social-emotional outcomes. Infants whose mothers presented more long-lasting depression symptoms (i.e., prenatally and postnatally depressed), those whose mothers showed more depression episodes, and those whose mothers had higher trait and state prenatal anxiety symptoms are expected to show more negative social-emotional outcomes than infants from the two other groups. Of note, the study was conducted on a low-risk sample that excluded confounding variables such as maternal low socioeconomic status, malnutrition, substance use problems, other psychiatric co-morbidities, significant environmental adversity, and neonatal complications (e.g., preterm delivery, low birth weight).

## 2. Materials & Methods

### 2.1. Design

This was a 20 month longitudinal cohort study that included several assessment points from the third trimester of pregnancy up to age 18 months (±1 month). Mothers were recruited in the prenatal care ward of the Paris University Hospital Pitié-Salpêtrière, France. The study protocol was approved by the Ethics Committee of Pitié-Salpêtrière Hospital, France (CCPRB N°19401, 13 March 2002).

### 2.2. Participants, Eligibility and Enrollment

Mothers were recruited at the beginning of the third trimester of pregnancy. They were included if they fulfilled the following criteria: fluency in French; age 20 to 38 years; primiparous. Exclusion criteria were: severe biomedical complications (acute or chronic physical diseases such as diabetes, metabolic diseases, hypertension, gestosis); multifetal pregnancies; signs of fetal malformation; drug/alcohol and cigarette (more than 15 per day) consumption; psychotic depression; other chronic psychiatric diseases of the expectant mother, except major depressive episodes without psychotic features; and after birth, prematurity. Of note, antidepressant or mood stabilizer medications are not recommended in France during pregnancy unless there is a history of psychotic depression during previous pregnancy. Because we included only primiparous women, we did not expect psychotropic medication. We hoped that this selection strategy would result in a community sample that had a relatively low risk of complications (during pregnancy, at birth, neonatal) and related stress.

A total of 205 mothers meeting the inclusion criteria were approached and screened. Fourteen of them were excluded due to one or more of the exclusion criteria. Twelve mothers refused to participate because of the length of the study (18 months). Data on reasons for non-inclusion were routinely collected for screened women fulfilling the inclusion criteria. There were no statistical differences between the inclusion and refusal groups in terms of age, socio-demographic status, or family status (single or in couple), except for cultural origin (French women represented 87.8% of mothers recruited and 58% of mothers who refused). After explanation of the research protocol, 164 mothers were then included. All of them provided written informed consent to participate in the study. Fathers were also informed and gave their written consent.

A full description of the study protocol and methods is available elsewhere [24]. The first interview conducted by research psychologists took place at the beginning of the third trimester of pregnancy at the maternity hospital (Figure 1). This included collection of socio-demographic information, screening measures with the Edinburgh Postnatal Depression Scale (EPDS) [25], the Center for Epidemiologic Studies Depression Scale (CES-D) [26], the Montgomery–Asberg Depression Rating Scale (MADRS) [27], systematic search for DSM-IV criteria of Major Depressive Episode (MDE), the Sensations during Pregnancy and Life Events Questionnaire (SENPLEQ) [28], the Symptom Check-List 90 Revised (SCL-90 R) [29] and the State-Trait Anxiety Inventory (STAI) [30]. Diagnosis of prenatal depression was based on positive MADRS score ≥ 15 and DSM-IV criteria of MDE confirmed by the Mini-International Neuropsychiatric Interview–MINI [31]. 

The second interview took place 3 days after delivery, at the maternity hospital, and was scheduled to evaluate the newborn’s neurobehavioral characteristics with the Neonatal Behavioral Assessment Scale (NBAS) [32] by a trained psychologist blind to the mother’s prenatal status. At 2 and 6 months postpartum, mothers replied to an interview (MADRS) conducted in the maternity hospital, with the same diagnostic criteria for depression as during pregnancy. At 12 and 18 months, the mothers were sent questionnaires at home on depression (EPDS) and anxiety symptoms (STAI), and on infant social-emotional development (Infant Toddler Social and Emotional Assessment—ITSEA) [33]. A clinically significant level of depressive symptoms was endorsed for an EPDS total score ≥ 12, as recommended in the original study of Cox et al. (1987) [25]. Data on maternal emotional states and on newborn and infant development at 1 year have been presented previously [24]. The present paper focuses on data from the 18 month assessment. 

### 2.3. Data Analysis

R software (version 3.3.1) was used for statistical computations, and a two-tailed *p* value < 0.05 was considered as significant for all analyses. In a first step, a descriptive analysis was performed: frequencies were computed for qualitative variables, and mean, standard deviation for quantitative variables. In a second step, a bivariate analysis was used based on a priori hypotheses. Given the non-normality of their distribution, infants’ ITSEA scores were compared across the three groups with a Kruskal–Wallis test. Associations between the number of maternal depression episodes and infants’ ITSEA scores, and between mothers’ trait and state anxiety scores and infants’ ITSEA scores were calculated with Kendall’s Tau correlation test. We used Holm correction to control for multiple comparisons. The low sample size in each group precluded performing analyses taking into account infants’ gender. 

## 3. Results

### 3.1. Trajectories of Maternal Depressive Symptoms

Mothers were screened for depression at five time points based on MADRS and/or EPDS assessments: 3rd trimester of the pregnancy, 2 months, 6 months, 12 months and 18 months postpartum. This resulted in 43 distinct trajectories comprising at least one participant. These trajectories were then merged into three major profiles: mothers who presented postnatal depression on at least one postpartum point without prenatal depressive symptoms (*n* = 19) (group DEP−/+); mothers who presented prenatal depression symptoms and postnatal depression on at least one postpartum point (group DEP+/+) (*n* = 14); and mother who never showed depression symptoms (group DEP−/−) (*n* = 38) (the study flow chart is presented in Appendix A). 

### 3.2. Socio-Demographic and Newborn Features between the Groups

Table 1 summarizes the socio-demographic and newborn characteristics of mothers included in the study, taking into account their depression status. Mothers came mostly from middle-to-upper socioeconomic levels. As expected, given exclusion criteria, the three groups did not differ in any of the assessed socio-demographic characteristics or in marital situation, maternal age, origin or weight gain during pregnancy. In addition, there were no differences in neonate birth weight and length. APGAR scores assessed 5 and 10 min after delivery showed good physiological condition for all neonates. 

### 3.3. Pre- and Postnatal Maternal Psychopathology

During the prenatal period, the EPDS total score, the CES-D total score and the MADRS total scores were significantly higher for mothers in the DEP+/+ group, followed by those in the DEP−/+ group, and by mothers in the DEP−/− group (Table 1). The three groups did not differ in terms of anxiety state, but trait anxiety was significant higher in the DEP+/+ mothers than in the DEP−/− mothers, while women in the DEP−/+ had an intermediate anxiety level. The number of stressful life events experienced was significantly higher in the DEP+/+ mothers than in the DEP−/+ mothers and in the DEP−/− mothers, but these differences were not significant after adjustment for multiple comparison testing. Mothers in the DEP+/+ group showed significantly higher reactivity to stress than those in the two other groups.

### 3.4. Association with Infant Social-Emotional Development

The total scores of internalizing symptoms of infants of women in the DEP+/+ group were higher than those of infants belonging to the two other groups, *H* (3) = 6.03, *p* = 0.049. In particular, the general anxiety index was higher in the DEP+/+ group compared to that in the two other groups, *H* (3) = 7.73, *p* = 0.021 (Table 2). Infants of mothers in the DEP−/− group had lower scores on the social relatedness scale than infants of mothers in the two other groups, *H* (3) = 6.65, *p* = 0.036.

### 3.5. Effect of the Duration of Maternal Depression on Infant Social-Emotional Development

A positive association was found between the number of maternal assessments with a clinically significant level of depressive symptoms and the score at the externalizing subscale of the ITSEA in infants at 18 months (*r_τ_* = 0.25, *p* = 0.013) (Table 3). Positive relationships were also found for the activity/impulsivity index and the aggression/defiance index. The relation between the number of maternal assessments with a clinically significant level of depressive symptoms and the score at the internalizing subscale of the ITSEA in infants at 18 months was not statistically significant (*r_τ_* = 0.13, *p* = 0.189). However, positive associations were found with the general anxiety index (*r_τ_* = 0.29, *p* = 0.006). 

### 3.6. Effect of Prenatal Anxiety on Infant Social-Emotional Development

Maternal state-anxiety during the 3rd trimester of the pregnancy was positively associated with externalizing symptoms of the ITSEA scores in infants at 18 months (*r_τ_* = 0.16, *p* = 0.029) and with two items of the dysregulation domain, i.e., negative emotionality (*r_τ_* = 0.15, *p* = 0.041) and eating (*r_τ_* = 0.16, *p* = 0.045) (Table 4). Maternal trait-anxiety was positively associated with internalizing symptoms of the ITSEA scores in infants at 18 months (*r_τ_* = 0.16, *p* = 0.032) and the aggression/defiance index (*r_τ_* = 0.19, *p* = 0.017), the compliance index (*r_τ_* = −0.20, *p* = 0.009) and the maladaptive index (*r_τ_* = 0.18, *p* = 0.021).

## 4. Discussion

### 4.1. Comments on the Main Findings

Infants whose mothers had depressive symptoms both in pre- and postnatal periods had more internalizing symptoms at 18 months compared to infants whose mothers had depressive symptoms only during the postnatal period or had no depressive symptoms. These infants were particularly at risk of reporting higher scores on the general anxiety and anxiety/worries index of the ITSEA. These findings are consistent with previous longitudinal studies reporting a positive link between antenatal depressive symptoms and negative effects in children, such as temperamental features in toddlers [34,35], internalizing problems in preschool children [36,37] or emotional and behavioral problems in 10–11 year old children [38].

Prior analyses of our data at the 12 month follow-up showed a comparable picture with higher scores on the generalized anxiety and activity/impulsivity index of the ITSEA in infants of prenatally depressed mothers at 1 year compared to women without perinatal depression [24]. The stability of the general anxiety index between 12 and 18 months is in line with the study conducted by McGrath, Records and Rice [35], in which temperamental traits relative to reactivity to distress in infants of antenatally depressed women were largely stable over the first months of life. 

Several hypotheses could be raised to explain the positive association between maternal antenatal depression and infants’ internalizing symptoms.

Firstly, maternal internalizing symptoms (i.e., depression/anxiety/stress) during pregnancy may affect infant development by altering the intrauterine environment and fetal maturation [5,6,7,8]. Animal models have provided an experimental framework to determine the biological pathways involved in this relationship. Dysregulation of the hypothalamic–pituitary–adrenal axis, a marker of stress reactivity, was regarded as the most plausible biological explanation for the negative impact of maternal internalizing symptoms during pregnancy on infants’ development [24,39]. The negative effect of persistently high levels of glucocorticoid during pregnancy was found both in animal models, where it induces anxiety-like behaviors [40,41], and in in vitro models, where it is associated with a reduction in proliferation of neural progenitor cells [42] or a specific expression of genes associated with high vulnerability to psychiatric disorders [43]. The implications of other biological pathways, such as changes in placenta function and inflammatory or epigenetic mechanisms, have been widely discussed elsewhere (for a review, see Fitzgerald, Parent, Kee and Meaney [44]). In addition, neuroimaging studies conducted in infants of women with antenatal mental health problems showed differences in brain structure and connectivity of regions specifically involved in threat assessment and cognitive appraisal [45,46]. Alterations in these structures may possibly serve an adaptative purpose in the postpartum period, helping neonates to monitor more effectively a less responsive interactive partner.

Secondly, the reported positive link between maternal antenatal depression and infants' internalizing symptoms at 18 months may reflect the effect of shared genetic and/or environmental risk factors. Reviews of psychosocial risk factors for antenatal depression reveal strong evidence for lack of social support along with presence of domestic violence, family stress, and low socioeconomic status [12] that were also associated with higher risk of impaired socio-emotional development in infants. We tried to moderate the risk of such confounding bias by including specifically mothers with low risk on family and medical status indicators. Data presented in Table 1 confirm that socio-demographic features and the number of stressful life events did not significantly differ across the groups, meaning that the contribution of those risk factors was limited in our sample.

Thirdly, antenatal depression is the strongest predictor of postnatal depression [12]. It is likely that the negative effects of antenatal depression observed here could be mediated by maladaptive parenting behaviors during the postpartum period in relation with maternal psychopathology [47]. In our research, all women who scored positively for antenatal depression had experienced depressive symptoms in the postnatal period. The analysis of the trajectories of maternal peripartum depressive symptoms showed that maternal depressive symptom levels were largely stable over the ante- and postnatal periods, in coherence with previous studies [11,12,13,14]. Said differently, a spontaneous remission of depressive symptoms is rarely observed in the postnatal period when symptoms previously occur during the pregnancy. We are aware that the division of our sample into three groups led to small sample sizes precluding the use of multivariate regression analyses. However, the very strict exclusion criteria made our groups rather homogeneous. The specific effect of antenatal depressive symptoms on child development, independent of maternal mental health status in the postpartum period or psychosocial risk factors, has also been confirmed in large community-based samples using multivariable models [16,37,48].

Of note, considering the close interplay between the emergence of cognitive function, inhibition abilities and emotional regulation from toddlerhood to school age [49], a high level of stress-related symptoms in infants of antenatally depressed women may result in later neurodevelopmental impairments. Much work stressed the negative effect of maternal antenatal depression on several neurodevelopment outcomes of children, such as early cognitive skills in toddlers (i.e., orientation skill, motor behavior, and self-regulation) [18], in executive function at 3 and 6 years [19] and several domains of general intelligence at 5 years [20].

The impact of combined pre- and postnatal depressive symptoms was only observed for internalizing symptoms, with marginal effects on other domains of the ITSEA scale (e.g., externalizing symptoms, dysregulation). These findings apparently contradicted previous findings showing the negative effect of peripartum depressive symptoms on externalizing symptoms in infants [50,51,52,53] but are consistent with other reports supporting a specific effect on internalizing symptoms [37,54]. Non-statistically significant findings in our study should be interpreted with caution, considering our low sample size. However, it is plausible that the other domains of psychopathology (e.g., externalizing behavior or sleep/food regulation) at 18 months were particularly under the influence of maternal psychopathology and concomitant ill-adapted parenting practices. In contrast, other mechanisms through which maternal depressive symptoms in the prenatal period contribute to infant psychopathology (e.g., shared genetic/environmental factors or direct effect on fetus development) may be especially important for internalizing symptoms. Interestingly, women in the DEP+/+ group had a higher anxiety trait, regarded as a psychological marker for anxiety. In the same way, subjects in the DEP+/+ group had higher reactivity to stressful life events compared to the other groups. Women with a combined form of peripartum depression (pre- and postnatal) may have higher vulnerability (either/both genetic or/and environmental) to internalizing problems compared to women who experienced depression only during the postpartum period, when several individual and environmental stressful factors influence maternal mood (e.g., hormonal, sleep deprivation, facing new parenting tasks, etc.).

A positive association was found between a proxy measure for the duration of depressive symptoms (i.e., the number of maternal assessments with clinically significant levels of depressive symptoms) and the scores of several externalizing and internalizing indexes of the ITSEA in infants at 18 months. These findings confirm the importance of taking into account the duration of maternal depressive symptoms with regard to the negative effect on children’s developmental outcomes [15,51,52]. For example, Kingston et al. (2018) [52] found that prolonged maternal depressive symptoms were associated with externalizing and internalizing children’s behaviors at age 3. The effects of ante- and postnatal depression were although not distinguished in these studies unlike the current research. 

### 4.2. Limitations

The specific characteristics of our population should be taken into account when discussing the generalizability of our results. Included mothers had low risk on family and medical status indicators that may not reflect the diversity of women with peripartum depressive symptoms. Additionally, attrition bias may have resulted in an overrepresentation of participants with high levels of depressive symptoms in our final sample. As discussed above, the sample was chosen to be the most homogeneous; however, no patient-centered method was applied to classify these women. Of note, the vast majority of latent class analyses and other forms of modeling using longitudinal cohort data applied in clinical or community-based samples differentiate distinct trajectories based on a gradient of severity of depressive symptoms (e.g., “no”, “low”, “moderate” and “high”) [19,20,37,50,51,52,54]. Here, we focused on the effect of the timing of depressive symptoms (i.e., ante vs. post) rather than their severity. In addition, the MADRS and the EDPS score were used to differentiate women with or without clinically significant depressive symptoms. Such a categorical approach may overlook women with a subthreshold depressive symptoms that could also negatively affect children’s development [55]. Maternal symptoms were self-reported, and infant psychopathology was parent-reported, which means that mothers completed both questionnaires in most cases. This could lead to measurement bias by artificially increasing the correlation between maternal and infant symptoms [56]. As depressed mothers are generally more prone to overrate externalizing symptoms and to underrate internalizing symptoms of their child [57], and as no significant association was reported between antenatal maternal depression and externalizing symptoms, it is unlikely that such informant bias influenced our principal findings. Further, for practical reasons, a self-reported measure of depression (i.e., EPDS) was used during the follow-up instead of the MADRS initially used at baseline. Finally, as noted above, a lack of statistical power due to a large number of variables and a relatively small sample size made it difficult to interpret non-significant results and precluded us from performing multivariate analyses.

## 5. Conclusions

Better recognition of the impact of antenatal depression, in addition to postpartum symptoms, is important to plan interventions, given that both periods are hormonally and psychologically different [55]. In practice, a high level of antenatal depression/anxiety/stress is a predictor of postnatal depression but is also likely to be associated with persistent internalizing symptoms in the newborn infant, which may interfere with their parent’s acquisition of parenting skills. Some aspects of parenting, such as the parent’s capacity to treat the child as a psychological agent or the capacity to provide contingent responsiveness towards the infant’s cues, have their roots in the antenatal period [58,59]. For example, maternal perception of fetal movement may represent a biological basis for maternal bonding during the antenatal period as a prerequisite for maternal–infant attachment in postpartum [60]. Prenatal depression can therefore be regarded as a psychiatric disorder with specific effects on infant development and as a marker of psychological vulnerability increasing the risk of emotional distress in the postpartum period for both the mother and her child. The effect may be stronger when additional biological and environmental factors occur [55].

As many women present transitory subclinical depressive symptoms in the immediate postpartum (often referred to as “baby blues”), some authors consider the screening of maternal emotional symptoms only in the immediate postnatal period as ill-adapted to discriminate women with potentially impairing depressive symptoms for their infant [14]. Interventions aimed at preventing adverse outcomes in the infant could be most effective if they begin early in the antenatal period, with repeated assessments during the postnatal period [61].

Clinical research is also needed to gain better knowledge of the features of maternal depressive symptoms associated with poor cognitive and affective outcomes in children, not only in terms of timing and duration, but also in terms of clinical phenomenology (e.g., anhedonia/irritability, associated anxiety). Future fundamental research could help in determining the possible biological pathways involved in the transmission of internalizing symptoms from pregnant mothers to toddlers. Such findings should offer more insight on the relevant targets for therapeutic and preventive interventions. 

## Figures and Tables

**Figure 1 jcm-11-06919-f001:**
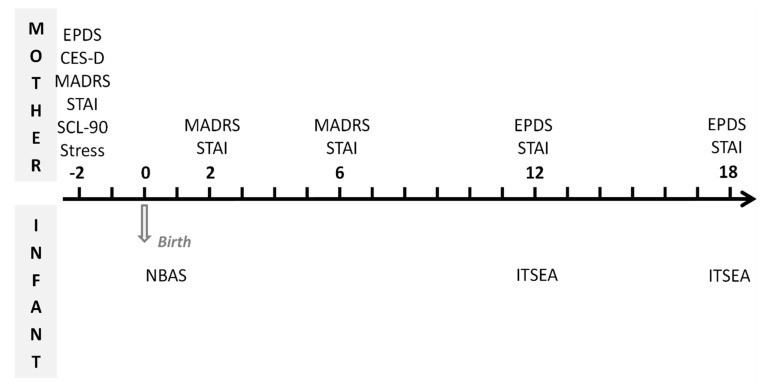
Measures and assessment points for mothers and infants.

**Table 1 jcm-11-06919-t001:** Socio-demographic, pregnancy and newborn characteristics and clinical features of mothers with no perinatal depression, postnatal depression only or prenatal and postnatal depression symptoms.

	Trajectories of Maternal Depressive Symptoms			Kruskal–Wallis Chi [2] or Fisher’s Exact	*p* Value	Adjusted *p*-Value (Holm)
	DEP−/− (No Perinatal Depression) *n* = 38	DEP−/+ (Postnatal Depression Only) *n* = 19	DEP+/+ (Prenatal and Postnatal Depression) *n* = 14			
Socio-demographic characteristics						
Mother’s age (years), mean ± SD	31.4 ± 4.0	30.8 ± 4.5	32.8 ± 4.0	1.69	0.430	1
Father’s age (years), mean ± SD	32.3 ± 5.3	32.8 ± 4.7	34.0 ± 7.10	0.10	0.949	1
Singles	14%	-	1 (8%)	-	-	1
Middle and low SES	49%	37%	54%	-	-	1
Pregnancy and newborn characteristics						
Weight gain (kg)	14.8 ± 3.9	14.1 ± 3.0	13.8 ± 5.3	1.04	0.596	1
Head circumference (cm)	34.7 ± 1.4	35.4 ± 4.2	35.4 ± 1.9	2.78	0.250	1
Weight (kg)	3.34 ± 0.49	3.18 ± 0. 35	3.61 ± 0.50	6.65	0.036 *	0.447
Length (cm)	49.3 ± 2.5	48.7 ± 1.9	50.5 ± 2.2	4.91	0.086	0.860
APGAR score 5′	10.0 (−)	9.9 ± 0.2	10 (−)	2.42	0.298	1
APGAR score 10′	9.8 ± 0.5	9.6 ± 1.0	9.9 ± 0.3	1.83	0.400	1
Prenatal maternal psychopathology						
MADRS score, prenatal	6.2 ± 4.4 ^c^	8.1 ± 4.0 ^b^	18.2 ± 3.6 ^a^	34.99	<0.001 **	<0.001 **
EPDS score, prenatal	3.4 ± 3.7 ^c^	7.3 ± 5.4 ^b^	10.6 ± 7.0 ^a^	15.31	<0.001 **	0.009 *
CES-D score, prenatal	8.7 ± 5.5 ^c^	13.3 ± 6.9 ^b^	24.2 ± 8.9 ^a^	21.45	<0.001 **	<0.001 **
STAI Y-A score, prenatal	28.8 ± 8.0	35.3 ± 12.9	38.7 ± 14.0	6.74	0.034 *	0.447
STAI Y-B score, prenatal	31.1 ± 8.8 ^c^	38.7 ± 9.9 ^b^	43.0 ± 9.3 ^a^	12.70	0.002 *	0.031 *
Number of SLE	7.8 ± 6.4 ^b^	9.4 ± 7.4 ^b^	19.0 ± 9.1 ^a^	8.04	0.018 *	0.269
Reactivity to SLE	21.3 ± 20.4 ^b^	29.2 ± 23.1 ^b^	61.6 ± 27.2 ^a^	12.33	0.002 **	0.036 *
Number of SPS	16.9 ± 10.0	15.0 ± 12.0	27.2 ± 17.5	2.72	0.256	1
Reactivity of SPS	40.7 ± 31.9	48.1 ± 44.1	94.0 ± 73.0	3.11	0.211	1
Postnatal maternal psychopathology						
MADRS score, 2 months	3.5 ± 3.3 ^b^	14.5 ± 9.3 ^a^	17.4 ± 8.8 ^a^	31.88	<0.001 **	<0.001 **
STAI Y-A, 2 months	27.1 ± 10.4	35.8 ± 15.5	32.4 ± 10.2	2.90	0.234	1
MADRS score, 6 months	3.0 ± 3.0 ^b^	13.0 ± 8.8 ^a^	12.8 ± 8.2 ^a^	28.02	<0.001 **	<0.001 **
STAI Y-A score, 6 months	28.3 ± 8.2	29.4 ± 8.2	37.2 ± 6.6	7.40	0.025 *	0.347
EPDS score, 12 months	3.4 ± 3.7	7.3 ± 5.4	10.8 ± 6.8	15.86	<0.001 **	0.008
STAI Y-A score, 12 months	48.9 ± 2.5	46.4 ± 4.1	48.1 ± 8.9	10.36	0.006 *	0.090
EPDS score, 18 months	2.9 ± 3.2 ^b^	7.6 ± 4.8 ^a^	6.9 ± 5.1 ^a^	14.34	0.001	0.015 *
STAI Y-A score, 18 months	49.4 ± 2.5	47.1 ± 3.8	48.9 ± 3.6	5.34	0.069	0.763

Note: EPDS = Edinburgh Postnatal Depression Scale; CES-D = Center for Epidemiologic Studies Depression Scale; MADRS = Montgomery–Asberg Depression Rating Scale; SES = socioeconomic status; SLE = Stressful Life Events; SPE = Stressful Pregnancy Sensations; STAI = State-Trait Anxiety Inventory. * indicates *p* < 0.05. ** indicates *p* < 0.01. ^a–c^ Means in a row without a common superscript letter differ (*p* < 0.05).

**Table 2 jcm-11-06919-t002:** ITSEA mean scores of infants of mothers with no perinatal depression, postnatal depression only, or prenatal and postnatal depression.

ITSEA Scores in Infants	Trajectories of Maternal Depressive Symptoms			Kruskal– Wallis	*p* Value
	DEP−/− (No Perinatal Depression) *n* = 38	DEP−/+ (Postnatal Depression Only) *n* = 19	DEP+/+ (Prenatal and Postnatal Depression) *n* = 14		
(1) Externalizing Symptoms					
Total score	0.36 ± 0.21	0.45 ± 0.28	0.54 ± 0.30	3.32	0.190
Activity/impulsivity	0.49 ± 0.38	0.70 ± 0.54	0.78 ± 0.39	5.12	0.077
Aggression/defiance	0.86 ± 0.46	0.93 ± 0.45	1.07 ± 0.54	1.16	0.559
Peer aggression	0.20 ± 0.23	0.15 ± 0.16	0.22 ± 0.23	0.66	0.719
(2) Internalizing Symptoms					
Total score	0.49 ± 0.21 ^b^	0.46 ± 0.24 ^b^	0.65 ± 0.18 ^a^	6.03	0.049 *
Depression/withdrawal	0.04 ± 0.06	0.06 ± 0.10	0.13 ± 0.19	1.96	0.376
Anxiety/worries	0.15 ± 0.20 ^c^	0.23 ± 0.17 ^b^	0.33 ± 0.21 ^a^	9.70	0.008 **
General anxiety	0.13 ± 0.17 ^c^	0.17 ± 0.11 ^b^	0.26 ± 0.16 ^a^	7.73	0.021 *
Separation distress	1.02 ± 0.42	0.93 ± 0.51	1.19 ± 0.43	2.62	0.270
Inhibition to novelty	0.76 ± 0.48	0.67 ± 0.44	1.02 ± 0.42	2.99	0.225
(3) Dysregulation					
Total score	0.37 ± 0.19	0.38 ± 0.22	0.52 ± 0.22	4.04	0.132
Sleep	0.43 ± 0.54	0.44 ± 0.50	0.56 ± 0.49	1.68	0.432
Negative emotionality	0.53 ± 0.31	0.54 ± 0.32	0.84 ± 0.44	4.60	0.100
Eating	0.17 ± 0.31	0.16 ± 0.31	0.22 ± 0.26	1.95	0.377
Sensory sensitivity	0.31 ± 0.22	0.34 ± 0.34	0.43 ± 0.26	1.62	0.446
(4) Competence					
Total score	1.11 ± 0.34	1.05 ± 0.23	0.97 ± 0.18	1.86	0.395
Compliance	1.06 ± 0.34	1.05 ± 0.34	0.87 ± 0.27	2.72	0.257
Attention	1.27 ± 0.46	1.26 ± 0.44	1.03 ± 0.52	1.99	0.370
Imitation/play	1.15 ± 0.50	1.07 ± 0.34	1.03 ± 0.53	0.21	0.900
Mastery motivation	1.41 ± 0.39	1.49 ± 0.36	1.39 ± 0.18	0.90	0.638
Empathy	0.82 ± 0.62	0.63 ± 0.48	0.66 ± 0.46	0.82	0.663
Prosocial peer relation	0.88 ± 0.46	0.66 ± 0.43	0.74 ± 0.46	2.25	0.325
(5) Additional Indices					
Maladaptive	0.15 ± 0.17	0.16 ± 0.16	0.16 ± 0.13	0.54	0.764
Social relatedness	1.70 ± 0.24 ^a^	1.59 ± 0.22 ^b^	1.52 ± 0.20 ^b^	6.65	0.036 *
Atypical	0.31 ± 0.27	0.34 ± 0.32	0.36 ± 0.31	0.24	0.887

Note: * indicates *p* < 0.05. ** indicates *p* < 0.01. ^a–c^ Means in a row without a common superscript letter differ (*p* < 0.05).

**Table 3 jcm-11-06919-t003:** Correlations between the number of maternal assessments with clinically significant level of depressive symptoms and infants’ ITSEA scores at 18 months.

ITSEA Scores in Infants	*n*	Kendall τ	Z	*p* Value
1. Externalizing Symptoms				
Total score	64	0.25	2.47	0.013 *
Activity/impulsivity	64	0.26	2.47	0.013 *
Aggression/defiance	64	0.26	2.28	0.023 *
Peer aggression	52	0.03	0.22	0.827
2. Internalizing Symptoms				
Total score	64	0.13	1.31	0.189
Depression/withdrawal	64	0.14	1.22	0.224
Anxiety/worries	64	0.34	3.19	0.001 **
General anxiety	64	0.29	2.76	0.006 **
Separation distress	64	0.10	0.94	0.348
Inhibition to novelty	64	0.03	0.29	0.772
3. Dysregulation				
Total score	64	0.15	1.49	0.136
Sleep	64	0.16	1.50	0.135
Negative emotionality	64	0.15	1.44	0.151
Eating	63	0.05	0.48	0.631
Sensory sensitivity	64	0.05	0.46	0.647
4. Competence				
Total score	64	−0.03	−0.35	0.725
Compliance	64	−0.07	−0.69	0.488
Attention	63	−0.11	−1.08	0.281
Imitation/play	64	0.05	0.49	0.622
Mastery motivation	64	0.10	0.94	0.347
Empathy	62	0.03	0.31	0.759
Prosocial peer relation	52	−0.08	−0.67	0.505
5. Additional Indices				
Maladaptive	64	0.11	1.05	0.294
Social relatedness	64	−0.19	−1.89	0.059
Atypical	63	0.03	0.32	0.752

Note: * indicates *p* < 0.05. ** indicates *p* < 0.01.

**Table 4 jcm-11-06919-t004:** Correlations between prenatal maternal anxiety and infants’ ITSEA scores at 18 months.

		STAI Y-B Score (Maternal Trait-Anxiety)			STAI Y-A Score (Maternal State-Anxiety)		
ITSEA Scores in Infants	*n*	Kendall τ	Z	*p* Value	Kendall τ	Z	*p* Value
1. Externalizing Symptoms							
Total score	91	0.11	1.57	0.117	0.16	2.18	0.029 *
Activity/impulsivity	91	0.11	1.42	0.156	0.17	2.28	0.023 *
Aggression/defiance	91	0.19	2.39	0.017 *	0.12	1.56	0.118
Peer aggression	72	−0.13	−1.42	0.155	−0.02	−0.19	0.852
2. Internalizing Symptoms							
Total score	91	0.16	2.15	0.032 *	0.04	0.54	0.592
Depression/withdrawal	91	0.09	1.06	0.288	0.08	0.95	0.343
Anxiety/worries	91	0.16	2.02	0.043 *	0.05	0.59	0.555
General anxiety	91	0.10	1.35	0.178	0.09	1.15	0.252
Separation distress	91	0.19	2.50	0.012 *	0.05	0.61	0.544
Inhibition to novelty	91	0.05	0.70	0.487	−0.03	−0.35	0.728
3. Dysregulation							
Total score	91	0.13	1.80	0.073	0.07	0.97	0.333
Sleep	91	0.03	0.41	0.679	−0.03	−0.41	0.680
Negative emotionality	91	0.11	1.47	0.142	0.15	2.04	0.041 *
Eating	91	0.15	1.85	0.064	0.16	2.00	0.045 *
Sensory sensitivity	91	0.13	1.73	0.084	0.02	0.21	0.831
4. Competence							
Total score	91	−0.03	−0.42	0.673	−0.14	−1.87	0.061
Compliance	91	−0.20	−2.62	0.009 *	−0.11	−1.53	0.126
Attention	90	−0.04	−0.51	0.609	−0.14	−1.88	0.060
Imitation/play	91	0.04	0.47	0.638	−0.08	−1.13	0.260
Mastery motivation	91	0.01	0.10	0.923	−0.07	−0.88	0.378
Empathy	90	0.04	0.52	0.601	−0.04	−0.55	0.579
Prosocial peer relation	73	−0.06	−0.71	0.477	−0.09	−1.09	0.275
5. Additional Indices							
Maladaptive	91	0.18	2.32	0.021 *	0.01	0.13	0.894
Social relatedness	91	0.01	0.17	0.866	−0.08	−1.01	0.311
Atypical	90	0.09	1.16	0.245	0.06	0.77	0.442

Note: * indicates *p* < 0.05.

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
