# Peer review of "Impact of the Timing of Maternal Peripartum Depression on Infant Social and Emotional Development at 18 Months"

_jcm, 2022, doi:10.3390/jcm11236919_

Round 1

Reviewer 1 Report

Thank you for the opportunity to review this article. The manuscript is a longitudinal cohort study assessing the link between internalizing and externalizing symptoms at 18 months using the ITSEA and maternal depression during pregnancy and up to four time points postpartum using the EPDS and MADRS. Previous results of the study population have been reported examining infant behavior at 12 months.  The results support other previous literature. They focused on low-risk pregnant women to hopefully reduce the confounders that are so closely intertwined with postpartum depression. Despite this, about 46.4% (DEP -/+ and DEP _+/+; 33/71) of women screened positive for perinatal depression. This exceeds the prevalence rate of one in ten women who report perinatal depression. This is an interesting finding that the authors do not report on and I would be curious to hear their thoughts on such a high number.

The screening measures they use are EPDS and MADRS, both of which have been valid and tested in a clinic and non-clinic population for screening. However, these are screening measures and are not diagnostic, rather they alert clinicians to the concern for perinatal depression. With the MADRS being a relatively low cut-off of >=15. Furthermore, the authors, switched from the MADRS to the EPSD at 6 mos postpartum. Was there a reason to use MADRS up to 6 mos postpartum and to switch to EPSD at 12 and 18 mos? Especially given that the EPSD is more sensitive to perinatal depression.

Overall, the finding supports the literature, but leads to a few more questions.

1)    Can you expound on the any key differences in results compared to your previous study at 12 mos that was published over 10 years ago? Why publishing this after 10 years from first finding? And would one expect to see different results at 12 mos vs 18 mos—what was the hypothesis?

2)    What about mothers with previous history of depression prior to pregnancy? Were these mothers screened out?

3)    Any treatment for the mothers who screened positive for perinatal depression?

4)    In table 1—it appears that the number of Stressful Life Events was statistically higher for the DEP +/+ group. 19.0 +/- 0.1 vs 9.4 for DEP-/+ with p value of 0.018. However, the authors say it was not statistically significant in the text of the paper (Lines 182-184). I would be curious to hear more about what type of SLE existed for these women.

5)    What was the EPDS cut-off?

Reviewer 2 Report

This paper is an extension of previous work by the same authors in that they are reporting on 18-month infant social-emotional outcomes as they are affected by maternal depression, looking at 3 different cohort groups of mothers (those with both pre- and postnatal depression, those with postnatal depression only, and those without depression). They have previously reported on their data in this longitudinal study at the infants’ 12 month mark. Key findings include that infants of mothers with prenatal and postnatal depression are more likely to have higher scores on the internalizing subscore of the ITSEA than infants whose mothers had only postnatal depression or no depression.

In general, the paper is well written , with the exception of several minor grammatical issues as follows.

Line 20:  “Assessment included the Edinburgh Postnatal Depression Scale, the State-Trait Anxiety  Inventory and the Infant Toddler Social and Emotional Assessment (ITSEA) at 18 months.” Although it is perhaps obvious which assessments were performed on mothers vs. infants, it would be more correct to say “Assessment of mothers included… while assessments of offspring included the ITSEA…”

Line 29: “These findings support the need of providing specific screening for women with prenatal depression.” Would be more correct to say: ”Specific screening to assess women for prenatal depression” or “specific screening to identify women with prenatal depression.”  (You don’t want to only screen women who already have prenatal depression, which is how your writing represents the situation.)

Line 60: “Anxiety has” not “anxiety have”

Line 128: “Diagnostic of…” should be “Diagnosis of…”

Line 211: “Was not statistically significance” should be “was not statistically significant.”

Line 229:  The use of “infants” instead of “offsprings” is welcomed.  While “offsprings” is technically accepted as a plural for “offspring” and used in some countries, at least in the US, “offspring” is noted to be used as either/and singular or plural status. However, the word “offspring” is not typically used to identify “infants” in a paper like this. Throughout this paper, I see uses of “offsprings,” “infants,” “neonates,” and “children.” It would be better to pick fewer words to describe the same thing. Typically, “infants” and “children” would be used.

Lines 250-251: “Hypothalamo-pituitary…” most often described as “hypothalamic-pituitary…”

Line 268: “Reviews of psychosocial risk factors for antenatal depression reveal strong evidence for social support, domestic violence, family stress, and low socio-269 economic status…”  I think this should be “strong evidence for lack of social support along with presence of domestic violence…” or “insufficient social support…”

Line 274:  Should be “the contribution…was limited…”

Lines 323-324: “These findings confirm the importance of taken into account…” should be “taking into account…”

Line 343: “on offsprings development…” should be “on offsprings’ development”

Line 358: “In practice, a high level of antenatal depression/anxiety/stress is a predictor of post-natal depression but is also likely to be associated with persistent internalizing symptoms in the neonatal period, with possible interferences with the acquisition of parenting skills.” This is a confusing sentence because it is unclear who has persistent internalizing symptoms; “in the neonatal period” suggests it’s the infant. But then it goes on to talk about acquisition of parenting skills, which has to refer to the parent, not the infant. The authors need to make clear that the first part (“neonatal period”) is referring to the infant. Possibly: “…persistent internalizing symptoms in the newborn infant, which may interfere with their parent’s acquisition of parenting skills.”

Line 380-381: “Fundamental research could help in future determining the possible biological pathways involved in the transmission of internalizing symptoms from pregnant mother to toddlers.” I would reword to either: “Future fundamental research could help in determining…” or “Fundamental research could help in future determination of the possible…”
